# Trend and decomposition analysis of factors influencing teenage pregnancy and motherhood in Nigeria, 2003–2018

Mobolaji M. Salawu[1], Rotimi Felix Afolabi[1]*, Ayo Stephen Adebowale[1,2], Martin Enock Palamuleni[2]

1 Department of Epidemiology and Medical Statistics, Faculty of Public Health, College of Medicine, University of Ibadan, Ibadan, Nigeria, 2 Population Studies and Demography Programme, Faculty of Humanities, North-West University, Mafikeng, South Africa

* rotimifelix@yahoo.com

## Abstract

### Background

Nigeria is among the countries with a high burden of Teenage Pregnancy and Motherhood (TPM) in sub-Saharan Africa. The adverse effect of TPM on young girls is enormous and often compromises their future socioeconomic advancement, including education. Limited number of studies have assessed the trends in TPM and the decomposition of its contributing factors. This study aimed to assess the levels, trends, and drivers of changes in TPM, between 2003 and 2018, in Nigeria.

### Methods

This study used a cross-sectional design with four consecutive rounds (2003, 2008, 2013, and 2018) of Nigeria Demographic and Health Survey datasets. Women aged 20–49 years who had ever terminated pregnancy, reported at least one childbirth or stillbirth before attaining age 20, were analysed. The outcome variable was having experienced TPM as a teenager. Data were analysed using trend and multivariate decomposition analyses at a 5% significance level.

### Results

The prevalence of TPM was 56.1%, ranging from 64.7% in 2003 to 55.7% in 2018. Overall, the prevalence of TPM decreased significantly by 10.7% over the studied period (p < 0.001). The change was due to a composite of a positive significant effect of the net compositional change (126%) and a negative effect of the net behavioural change (26%). The identified significant drivers of shift in TPM due to changes in the composition of women included current age, educational level, employment status, timing of marriage, age at first sexual intercourse, contraceptive use, ethnicity, and

**Data availability statement:** https://dhspro-gram.com/data/available-datasets.cfm

**Funding:** The author(s) received no specific funding for this work.

**Competing interests:** The authors have declared that no competing interests exist.

region of residence. Due to the change in behaviour, TPM reduced by 20% among South-South residents compared with their North-Central counterparts. However, TPM increased by 260% among teens who had their first sexual initiation.

## Conclusions

The TPM prevalence remained high in Nigeria, though a decreasing trend was observed within the studied period. Government and other stakeholders should focus pragmatic interventions on the identified drivers of TPM change over the last two decades in their efforts to alleviate TPM in Nigeria.

## Introduction

Globally, teenage pregnancy and motherhood (TPM) is a public health challenge with associated poor obstetrics and perinatal outcomes, including future socio-economic challenges [1,2]. The teenage period is a transitional phase between the end of childhood and shortly after the beginning of adolescence, which stretches from thirteen to nineteen years [2]. In this study, TPM refers to the experience of all forms of pregnancy endpoints including childbirth among teenagers who ever conceived. The prevalence of TPM is declining worldwide. However, enormous variations still exist in Low- and Middle-Income Countries (LMICs), particularly in sub-Saharan Africa [2]. Of 21 million girls aged 15–19 years who become pregnant every year in LMICs, 12 million give birth – of which girls aged 15 years and below account for two million births [3,4].

The TPM is still prevalent in Africa, particularly in West Africa. The top six countries with the highest teenage pregnancy rate are Niger Republic (203/1000 women aged 15–19 years), Mali (175), Angola (166), Mozambique (142), Guinea (141), and Chad (137), while Nigeria (111) ranks in the sixteenth position [5–7]. Twenty percent of teen girls in Nigeria are already mothers or pregnant, ranging from 2% (aged 15) to 37% (aged 19). The pregnancy- and childbirth-related conditions have contributed to poor infant health outcomes and high maternal morbidity and mortality, of which Nigeria contributed about 28.5% of global maternal mortality [8].

Teenage pregnancy and motherhood adversely impact the health and well-being of the girl child. The health concerns and impact of TPM on teenagers are enormous and challenging. These challenges include various socio-economic issues and obstetric complications that can negatively affect future reproductive health. Sexually transmitted infections and other systemic infections are common complications associated with TPM [2,9]. Additional adverse effects include anaemia and sepsis, both of which can lead to death. Also, the under-developed pelvis of teenage girls often leads to cephalopelvic disproportion, resulting in obstruction during childbirth and subsequent obstetric fistula [10,11].

Other complications include premature childbirth, which is linked to poor perinatal outcomes such as preterm births, intrauterine growth restrictions, and low birth weight. These complications increase the risk of infections, malnutrition, other

morbidities and even mortality among infants [11,12]. Maternal death is another adverse outcome associated with TPM. According to WHO, maternal conditions are the second leading cause of death globally among girls aged 15–19 [4,13]. Furthermore, socio-economic challenges are inevitable and multifaceted, including academic difficulties leading to school drop-out, financial hardships, inadequate nutrition, and the emotional burden of shame and stigma often experienced by teen mothers [11,14].

Several factors have been reported to have interaction and influence on the occurrence of TPM in Nigeria. Some of the reported factors are socio-economic and cultural, such as educational level, age, marital status, religion, poverty, income level, and occupation [15,16]. For instance, child marriage remains a cultural practice in northern Nigeria. Despite existing policies aimed at curbing child marriage, early sexual debut is common among adolescent girls across Nigeria [17]. Others are multifaceted factors which include peer pressure, poor contraceptive use, early menarche, child marriage, sexual violence, place of residence, geopolitical zone, gender inequality, exposure to and use of social media/internet, psychological and other health-related factors [18–20].

Teenage pregnancy and motherhood remain prevalent in Nigeria despite various interventions and research efforts. The prevailing level of TPM in Nigeria is among the major contributory factors to a high fertility, which has been persistently reported in the literature [19]. Empirical evidence has shown that Nigeria is among the countries identified for targeted interventions to reduce the global population growth rate [21]. Despite the importance of TPM in Nigeria to the global fertility and maternal mortality dynamics, limited research has been conducted to assess the sequence, course, and drivers of this demographic issue over time [17,22,23].

Most cross-sectional studies [5,6,9,17,23] have been conducted among current teenagers, substantially underestimating the burden of TPM in Nigeria and elsewhere. This approach fails to account for individuals who experienced TPM in their teens but are now adults, potentially overlooking the true extent of the issue and undermining effective intervention strategies [22,24]. Additionally, studies have assessed trends and investigated factors associated with teen pregnancy and childbearing. However, little is known about the changes in TPM and the drivers of these changes over time in Nigeria.

Therefore, the study was conducted to assess the trends in TPM and examine the drivers of changes in TPM among women aged 20 years and above in Nigeria. The present study focused on women who have completed their teenage years, considering all pregnancy endpoints – childbirth, miscarriage, or abortion, and employed multivariate decomposition analysis (MDA) to account for the changes in TPM across the studied period. The outcome will help document factors contributing to changes in TPM within the last two decades, which is instrumental in designing relevant policy measures. It could also provide information for adolescent health care services, and for preventing morbidity and mortality associated with TPM, guiding efforts towards achieving the Sustainable Development Goals in Nigeria and worldwide [25].

## Methods

### Study area

The study was conducted in Nigeria, the most populous country in Africa. Nigeria has 36 states and the Federal Capital Territory as its administrative area, with a projected population of 223.8 million in 2023 [26]. There are six geopolitical regions in Nigeria, each of which comprises at least five (5) administrative areas, officially referred to as the 'States'. The country has multi-ethnic groups, but the predominant ethnic groups are Hausa, Fulani, Igbo, and Yoruba.

### Data source and study design

This cross-sectional study design utilised four consecutive rounds of NDHS conducted in 2003, 2008, 2013, and 2018. These nationally representative surveys were designed to provide information on population and fertility-related behaviours of women of reproductive age in Nigeria. Two-stage stratified cluster design was used based on the adopted sampling frame from the 1991 census for the 2003 NDHS and from the 2006 Nigeria population and housing census

for 2008, 2013, and 2018 NDHSs. The primary sampling units, referred to as clusters, were the enumeration areas. In total, 7,985, 36,800, 40,680, and 42,000 households were sampled from 365 (165 urban, 200 rural), 888 (286 urban, 602 rural), 904 (372 urban, 532 rural), and 1,400 (372 urban, 532 rural) clusters selected for the 2003, 2008, 2013, and 2018 NDHSs, respectively. Well-trained interviewers administered the pre-tested, semi-structured questionnaire to the individual women during face-to-face interviews. Detailed reports of the sample design and sampling procedures have early been published [8,27–29].

## Study population

A total of 7,620, 33,385, 38,948, and 41,821 women participated in the surveys held in 2003, 2008, 2013, and 2018, respectively. In this study, 93,201 women were included, consisting of 5,567 (2003), 25,677 (2008), 29,838 (2013), and 32,119 (2018) women who had attained age 20 years as of the respective date of the surveys. These were the participants who had reported at least one childbirth, stillbirth or pregnancy termination, and supplied all required dates for relevant events as of the survey date.

## Study variables

**Outcome variable.** The outcome variable was teenage pregnancy and motherhood (TPM). The TPM was defined as conception by a woman before attaining age 20 years [22,30], irrespective of whether the pregnancy resulted in childbirth or was voluntarily or involuntarily terminated. Many studies have estimated TPM using numbers of women who were either "currently pregnant" or "ever-had-a-childbirth" as teenagers [5,17,23]. These studies fail to account for those who experienced TPM as teenagers but are now adults, thereby underestimating its true burden [22,24]. The present study therefore considered women who had already completed their teen years using retrospective information about the timing of their first pregnancy and childbirth. The following variables/questions: "age of respondent at first childbirth", and "have you ever had a pregnancy that was miscarriage, aborted, or ended in a stillbirth?" and "when did the last such pregnancy end?" were used to determine respondents who had reported pregnancy or childbirth before, and those who had pregnancy terminated before age 20 years. The responses considered were dichotomised as "ever experienced TPM" and "never experienced TPM"; these were coded as "1" and "0", respectively. Although some teenagers who reported the exact age of 20 years to any of the three variables might have been pregnant before or shortly after age 20 years, this would have minimal effect on our findings. For this study, teenagers are therefore women who had their first pregnancy or childbirth from age 13–19 years.

**Explanatory variables.** The explanatory variables included in this study and their respective classification are as follows.

1. Demographic and socio-economic factors: age (20–24, 25–34, ≥ 35 years), education (no formal, primary, and post-primary), employment (not working, currently working), household wealth status (poor, average, rich), and media exposure (none, exposed)

2. Cultural factors: ethnicity (Hausa/Fulani, Igbo, Yoruba, and Others) and religion (Christian, Muslim and Others)

3. Fertility indicators: timing of marriage (unmarried at 20, early-teen (< 15 years), mid-teen (15–17 years), and late-teen (18–19 years)), age at first sex (<15, 15–19, > 19 years), and contraceptive use (never, ever)

4. Community factors: residence (urban, rural); region of residence (North-Central, North-East, North-West, South-East, South-South, and South-West), and community social economic disadvantage (least, low, middle, high, most).

In this study, the media exposure variable was generated and recoded as 1 "exposed" if a household had access to at least radio, television, or newspaper; otherwise, 0 "none". The wealth index variable was derived from the generated

weighted factor score using principal component analysis. These scores were coded as 0 "poor", 1 "average" and 2 "rich" wealth tertiles. Additionally, the community socio-economic status (SES) index was derived from the proportion of women, within the same clusters, without education, belonging to a household in the two lowest wealth quintiles, having none media exposure, and unemployed using the principal component factor method as used in earlier studies [31,32]. The four variables, represented as proportion, were standardised to ensure that those with higher prevalence did not disproportionately influence the overall index. Principal component analysis was thereafter applied to these variables to determine their respective variable loadings on the first principal component. The first principal component score for each cluster was computed by multiplying each standardised variable by its respective weight, and summing the results across all variables. The scores were coded as 0 "least", 1 "low", 2 "middle", 3 "high" and 4 "most" disadvantaged community SES quintiles.

## Statistical methods of analysis

Data were weighted before use due to the complex sampling design used during data collection to adjust for differences in the population sizes of each region in Nigeria. At the univariate level, descriptive statistics were used to describe TPM by the selected explanatory variables. Specifically, the frequency distribution and prevalence of TPM, including its percentage changes by the explanatory characteristics, were presented. Across the periods between 2003 and 2018, trends in TPM for 2003–2008, 2008–2013, 2013–2018, and 2003–2018 time points were evaluated. The chi-square test for trends of proportions was used to identify the significant changes across the time points. The MDA was thereafter employed to decompose changes in TPM between 2003 and 2018.

**Multivariate decomposition analysis (MDA).** The MDA utilises models, including logistic regression, to quantify how explanatory variables contribute to differences in the probability of events occurring across mutually exclusive groups. This approach contrasts with the standard logistic regression, which identifies the odds of an event occurring. In MDA, the difference in women having experienced TPM was the outcome variable. The outcome in 2003 constituted a 'group' and that of 2018 constituted another 'group', while predictor effects were separated into differences in characteristics (or endowment) and differences in the effects (or coefficients) in the regression decomposition [32,33]. This approach allows the root of changes in TPM between 2003 and 2018 to be identified, and evaluates how changes in TPM were impacted by the selected explanatory characteristics.

In this study, the decomposition of the difference in the factors influencing TPM is a function of a linear combination of the predictors and regression coefficients. Generally, the function can be additively decomposed into:

$$Y = F(X\beta) \tag{1}$$

where $Y - n \times 1$ vector of the coefficients of the outcome variable, $X - n \times p$ matrices of the explanatory variables, and β is the $- p \times 1$ vector of the regression coefficients in Equation 1

The difference in the proportion of respondents with TPM was decomposed into two parts, as expressed in Equation 2.

$$Y_v - Y_{1-v} = F(X_v \beta_v) - F(X_{1-v} \beta_{1-v}) \tag{2}$$

Equation 2 is further expressed in terms of explained and unexplained components in Equation 3 as follows:

$$Y_v - Y_{1-v} = [F(X_v \beta_v) - F(X_{1-v} \beta_v)] + [F(X_{1-v} \beta_v) - F(X_{1-v} \beta_{1-v})] \tag{3}$$

The first part of Equation 3 represents the differential attributable to differences in endowment (explained component), while the other component is the differential attributable to differences in coefficients (unexplained component). Additionally, $Y_v$ represents the proportion of respondents who had experienced TPM (comparison group), while $Y_{1-v}$ represents

the proportion of respondents without TPM experience (reference group), and $0 \leq v \leq 1$. This method has been used in another study [22,34]. Data management and analysis were conducted using STATA 17 SE (StataCorp, College Station, Texas, United States of America). Multicollinearity was assessed, and the variance inflation factor (VIF) was evaluated. All analyses were carried out at a 5% level of significance.

## Ethical approval

Ethical approval for the parent study was obtained by Measure Demographic and Health Survey (DHS) from the National Health Research Ethics Committee (NHREC) and the ICF Institutional Review Board. Measure DHS reported the details of the ethical approval [8,27–29]. This study utilised a secondary dataset, freely available for use in the public domain, which requires no ethics approvals. Meanwhile, the DHS program and ICF Macro USA granted authorisation to access the raw data set used for the present analysis.

## Results

### Women characteristics

The distribution of the women characteristics is presented in Table 1. Women's mean age was 32.3 ± 8.3, ranging from 31.8 ± 8.4 in 2003 to 32.6 ± 8.3 in 2018. Most women were aged ≥25 years (80.4%), lived in a rural area (59.7%), were from the North-west (28.5%), and were of Hausa/Fulani ethnicity (35.7%). About two-fifths of the women had no formal education (40.5%) and were from poor households (38.2%). More than one-fifth of the women had first sexual intercourse at age < 15 years, and two-fifths were unmarried before age 20. Only 33.0% had ever used contraceptives, ranging from 29.2% in 2013 to 36.2% in 2003.

### Trends and bivariate analysis of TPM in Nigeria

Table 2 shows the prevalence, percentage change, and significance of changes in having experienced TPM by the selected women characteristics. With an overall prevalence of 56.1% between 2003 and 2018, the prevalence rates of having experienced TPM in 2003, 2008, 2013, and 2018 were 62.4, 54.2, 57.0 and 55.7%, respectively (Fig 1).

On average, between 2003 and 2018, TPM was more prevalent among women from a poor wealth household (72.5%) compared with those from the rich (39.1%), among Muslims (70.4%) versus Christians (40.8%), among uneducated women (74.9%) compared with women with post-primary education (33.7%), among rural (64.9%) versus urban (43.1%) residents, among women who had an early-teen (86.0%) or mid-teen (88.6%) marriage compared with those who were unmarried before age 20 (12.6%). Most women who had experienced TPM lived in a community with the most disadvantaged socio-economic status (71.0%), and had never used contraceptives (62.0%) or had sexual intercourse before age 15 (82.5%) (Table 2).

In Table 2, the prevalence of having experienced TPM reduced by 13.3% between 2003 and 2008, increased by 5.4% between 2008 and 2013, and decreased by 2.3% between 2013 and 2018. Overall, TPM prevalence decreased by 10.7% between 2003 and 2018. These reductions were significant (p < 0.001) over the study period. Trends in having experienced TPM were significant (p < 0.05) for most women characteristics, except among women who had primary or post-primary education, women aged 25–34 years, practiced 'others" religion, married either at age 18–19 years, ever used contraceptives, lived in the North East or the community with the least disadvantage socio-economic status. Noteworthy, a significant increasing trend in having experienced TPM was observed among women who were aged 20–24 years, unexposed to media, not working, or Hausas/Fulanis, had an early-teen/ mid-teen marriage, sexual intercourse at age < 15 years, or no formal education, and resided in poor household wealth, North East or in a community with low, middle, high or most disadvantage socio-economic status. Specifically, the highest increased trend of having experienced TPM was reported among women who lived in the most disadvantaged socio-economic status (20.8%).

**Table 1. Distribution of respondents according to background characteristics by survey year.**

| Background Characteristics | 2003 n^ (%^^) | 2008 n^ (%^^) | 2013 n^ (%^^) | 2018 n^ (%^^) | Overall n^ (%^^) |
|---|---|---|---|---|---|
| **Age (year)** | | | | | |
| 20-24 | 1214 (22.9) | 5271 (20.5) | 5813 (19.6) | 5884 (18.3) | 18182 (19.6) |
| 25-34 | 2251 (40.3) | 10599 (41.5) | 12131 (41.2) | 12918 (41.0) | 37899 (41.1) |
| ≥35 | 2102 (36.8) | 9807 (38.1) | 11894 (39.2) | 13317 (40.7) | 37120 (39.3) |
| *mean ±sd* | *31.8±8.4* | *31.9±8.3* | *32.3±8.4* | *32.6±8.3* | *32.3±8.3* |
| **Education** | | | | | |
| No formal | 2529 (47.0) | 11394 (40.0) | 11770 (41.8) | 12229 (38.4) | 37922 (40.5) |
| Primary | 1243 (21.8) | 5482 (21.5) | 6040 (19.1) | 5416 (15.9) | 18181 (18.8) |
| Post-primary | 1795 (31.2) | 8801 (38.4) | 12028 (39.2) | 14474 (45.7) | 37098 (40.7) |
| **Employment** | | | | | |
| Not working | 1845 (33.7) | 8461 (31.7) | 8363 (28.5) | 8669 (26.6) | 27338 (29.0) |
| Currently working | 3722 (66.3) | 17216 (68.3) | 21475 (71.5) | 23450 (73.4) | 65863 (71.0) |
| **Household wealth** | | | | | |
| Poor | 2209 (39.6) | 11206 (38.7) | 11233 (38.5) | 12604 (37.2) | 37252 (38.2) |
| Average | 1058 (19.2) | 4983 (18.6) | 5951 (18.7) | 6700 (19.4) | 18692 (18.9) |
| Rich | 2300 (41.2) | 9488 (42.7) | 12654 (42.8) | 12815 (43.4) | 37257 (42.9) |
| **Ethnicity** | | | | | |
| Hausa/Fulani | 2049 (40.9) | 8407 (31.9) | 9683 (36.2) | 10992 (37.4) | 31131 (35.7) |
| Igbo | 910 (12.5) | 3313 (14.9) | 4058 (14.1) | 5146 (15.5) | 13427 (14.7) |
| Yoruba | 761 (11.3) | 3676 (17.6) | 4547 (15.4) | 4238 (15.8) | 13222 (15.9) |
| Others | 1847 (35.2) | 10281 (35.6) | 11550 (34.2) | 11743 (31.3) | 35421 (33.6) |
| **Religion** | | | | | |
| Christian | 2674 (45.0) | 12699 (51.7) | 14863 (46.1) | 15936 (46.8) | 46172 (47.8) |
| Muslim | 2774 (53.3) | 12329 (46.1) | 14532 (52.4) | 15892 (52.6) | 45527 (50.8) |
| Others | 119 (1.6) | 649 (2.2) | 443 (1.5) | 291 (0.6) | 1502 (1.4) |
| **Timing of marriage (year)** | | | | | |
| Unmarried at 20 | 1855 (30.2) | 9350 (39.5) | 11463 (37.4) | 13517 (42.3) | 36185 (39.2) |
| Early-teen (13–14) | 1662 (32.0) | 6158 (22.9) | 6722 (23.2) | 5874 (18.6) | 20416 (22.1) |
| Mid-teen (15–17) | 1365 (25.5) | 6992 (25.4) | 7777 (26.4) | 8453 (26.2) | 24587 (26.0) |
| Late-teen (18–19) | 685 (11.7) | 3177 (12.2) | 3876 (13.0) | 4275 (12.9) | 12013 (12.7) |
| **Age at first sex (year)** | | | | | |
| <15 | 1689 (32.9) | 5655 (21.1) | 6995 (24.0) | 6002 (18.7) | 20341 (21.9) |
| 15-19 | 2768 (50.0) | 14016 (53.9) | 15335 (51.2) | 19418 (59.9) | 51537 (54.9) |
| ≥20 | 1110 (17.1) | 6006 (25.0) | 7508 (24.8) | 6699 (21.4) | 21323 (23.2) |
| **Contraceptive use** | | | | | |
| Never | 3561 (63.8) | 17880 (65.4) | 21082 (70.8) | 21538 (65.4) | 64061 (67.0) |
| Ever | 2006 (36.2) | 7797 (34.6) | 8756 (29.2) | 10581 (34.6) | 29140 (33.0) |
| **Media Exposure** | | | | | |
| None | 1203 (22.6) | 5449 (18.9) | 6626 (21.3) | 9379 (27.4) | 22657 (22.8) |
| Exposed | 4364 (77.4) | 20228 (81.1) | 23212 (78.7) | 22740 (72.6) | 70544 (77.2) |
| **Place of Residence** | | | | | |
| Urban | 2217 (34.2) | 7898 (35.1) | 11592 (41.0) | 12743 (44.9) | 34450 (40.3) |
| Rural | 3350 (65.8) | 17779 (64.9) | 18246 (59.0) | 19376 (55.1) | 58751 (59.7) |
| **Region** | | | | | |
| North Central | 944 (14.6) | 4801 (13.9) | 4709 (14.0) | 5932 (14.1) | 16386 (14.1) |

*(Continued)*

**Table 1.** (Continued)

| Background Characteristics | 2003<br>n^ (%^^) | 2008<br>n^ (%^^) | 2013<br>n^ (%^^) | 2018<br>n^ (%^^) | Overall<br>n^ (%^^) |
|---|---|---|---|---|---|
| North East | 1079 (18.5) | 4861 (13.0) | 5167 (14.8) | 5761 (15.4) | 16868 (14.7) |
| North West | 1386 (29.4) | 5936 (25.3) | 7610 (31.0) | 7606 (28.6) | 22538 (28.5) |
| South East | 690 (8.8) | 2602 (11.3) | 3267 (10.9) | 4249 (11.9) | 10808 (11.2) |
| South South | 663 (16.5) | 3655 (16.3) | 4549 (12.6) | 4048 (12.0) | 12915 (13.6) |
| South West | 805 (12.1) | 3822 (20.2) | 4536 (16.6) | 4523 (18.1) | 13686 (17.8) |
| **Community SES** | | | | | |
| Least disadvantage | 32 (1.4) | 3645 (20.0) | 4457 (15.9) | 10501 (37.5) | 18635 (23.6) |
| Low disadvantage | 803 (13.8) | 5156 (20.6) | 6107 (21.2) | 6534 (21.2) | 18600 (20.6) |
| Middle disadvantage | 1594 (28.6) | 5594 (20.9) | 6006 (19.0) | 5486 (15.7) | 18680 (19.0) |
| High disadvantage | 1674 (28.2) | 5749 (20.3) | 6394 (20.4) | 4826 (13.2) | 18643 (18.4) |
| Most disadvantage | 1464 (27.9) | 5533 (18.1) | 6874 (23.5) | 4772 (12.4) | 18643 (18.5) |
| *Total* | *5567* | *25677* | *29838* | *32119* | *93201* |

n –number of women per group;

^unweighted frequency;

^^weighted percentage; sd - standard deviation, SES – socio-economic status

### Multivariate decomposition of changes in TPM between 2003 and 2018

Table 3 presents the effect of the selected women characteristics on changes in TPM level between 2003 and 2018. Specifically, the table reveals how much of the difference is attributable to changes in women characteristics (endowments/explained component), and how much to the effects of these characteristics (coefficients/unexplained component). During the studied period, the decomposition of the changes in having had TPM experience between 2003–2018 revealed significant disparity (coef: 0.0837; p-value<0.001), and about 126% of the reduction in prevalence of TPM was linked to the differences in characteristics or endowments. However, overall change attributed to the difference in coefficients explained about 26% nonsignificant increase in the prevalence of TPM over the studied period. If the behaviour of the women population had remained the same between 2003 and 2018, the TPM level would have been 26% higher. Overall, the result demonstrates that the reduction in TPM was largely due to changes in the compositional differences compared to behavioural changes among the respondents; further buttressing the overall decrease explained by the endowments.

The most significant contributors to the decrease in TPM were attributed to the differences in the composition of women age (5.6%), education (11.1%), timing of marriage (134.6%), contraceptive use (1.3%), ethnicity (6.9%) and region (3.1%). In particular, the change in the composition of women who married at early-teen (150.6%) showed the highest significant contribution to decreasing TPM prevalence. Conversely, the change in the composition of women who were unemployed and had sexual intercourse at age 15–19 years significantly contributed 5% and 27%, respectively, to the increase in TPM. The compositional effect of age at sexual initiation was positive for those who had sexual initiation before age 15 years, and negative for those who did between age 15–19 years. Having controlled for the effect of compositional factors, change in the behaviour of women who were married at early-teen and those who resided in the South-South revealed about a 45% and 19% significant reduction in the prevalence of TPM, respectively. Conversely, the change in behaviour among women engaging in teen sexual intercourse showed a significant increase in TPM (Table 3).

## Discussion

Teenage pregnancy and motherhood (TPM) constitute a major public health issue with negative health and socio-economic impact on the young mother, the newborn, her family, and society at large. An insight into the drivers of TPM in the last two decades will aid in designing a policy framework for the improvement of the health and well-being

**Table 2. Prevalence and trend of TPM by women background characteristics according to the survey year.**

| Background Characteristics | Prevalence | | | | | Percent changes | | | | |
|---|---|---|---|---|---|---|---|---|---|---|
| | 2003 | 2008 | 2013 | 2018 | Overall | 2003-2008 | 2008-2013 | 2013-2018 | 2003-2018 | p-value |
| **Age (year)** | | | | | | | | | | |
| 20-24 | 61.9 | 57.0 | 61.8 | 63.0 | 60.8 | −7.9 | 8.4 | 1.9 | *1.7 | <0.001 |
| 25-34 | 59.1 | 51.8 | 55.2 | 54.4 | 54.2 | −12.4 | 6.6 | −1.4 | −8.6 | 0.329 |
| ≥35 | 66.2 | 55.2 | 56.5 | 53.8 | 55.7 | −16.6 | 2.4 | −4.8 | −23.0 | <0.001 |
| **Education** | | | | | | | | | | |
| No education | 75.5 | 71.6 | 74.3 | 78.1 | 74.9 | −5.2 | 3.8 | 5.1 | *3.4 | <0.001 |
| Primary | 68.4 | 62.6 | 64.2 | 64.8 | 64.2 | −8.5 | 2.6 | 0.9 | −5.3 | 0.498 |
| Post-primary | 38.4 | 31.2 | 35.0 | 33.8 | 33.7 | −18.8 | 12.2 | −3.4 | −12.0 | 0.197 |
| **Employment** | | | | | | | | | | |
| Not working | 61.6 | 54.5 | 57.0 | 63.1 | 58.5 | −11.5 | 4.6 | 10.7 | *2.4 | <0.001 |
| Currently working | 62.8 | 54.0 | 57.0 | 53.1 | 55.1 | −14.0 | 5.6 | −6.8 | −15.4 | <0.001 |
| **Wealth Index** | | | | | | | | | | |
| Poor | 72.4 | 69.1 | 72.9 | 75.1 | 72.5 | −4.6 | 5.5 | 3.0 | *3.7 | <0.001 |
| Average | 70.5 | 59.3 | 61.8 | 61.4 | 61.5 | −15.9 | 4.2 | −0.6 | −12.9 | <0.001 |
| Rich | 48.9 | 38.4 | 40.6 | 36.6 | 39.1 | −21.5 | 5.7 | −9.9 | −25.2 | <0.001 |
| **Ethnicity** | | | | | | | | | | |
| Hausa/Fulani | 77.3 | 73.7 | 76.1 | 79.5 | 76.8 | −4.7 | 3.3 | 4.5 | *2.8 | <0.001 |
| Igbo | 36.9 | 31.0 | 33.2 | 28.8 | 31.2 | −16.0 | 7.1 | −13.3 | −22.0 | <0.001 |
| Yoruba | 38.4 | 36.6 | 39.1 | 33.6 | 36.4 | −4.7 | 6.8 | −14.1 | −12.5 | 0.016 |
| Others | 61.8 | 55.0 | 54.8 | 51.7 | 54.3 | −11.0 | −0.4 | −5.7 | −16.3 | <0.001 |
| **Religion** | | | | | | | | | | |
| Christian | 49.7 | 41.8 | 41.7 | 37.4 | 40.8 | −15.9 | −0.2 | −10.3 | −24.7 | <0.001 |
| Muslim | 73.0 | 67.7 | 70.3 | 72.0 | 70.4 | −7.3 | 3.8 | 2.4 | −1.4 | <0.001 |
| Others | 65.7 | 60.5 | 64.1 | 60.8 | 62.2 | −7.9 | 6.0 | −5.1 | −7.5 | 0.416 |
| **Timing of Marriage (year)** | | | | | | | | | | |
| Unmarried at 20 | 14.9 | 12.2 | 11.2 | 13.7 | 12.6 | −18.1 | −8.2 | 22.3 | −8.1 | 0.005 |
| Early-teen (13–14) | 85.6 | 81.5 | 87.4 | 88.8 | 86.0 | −4.8 | 7.2 | 1.6 | *3.7 | <0.001 |
| Mid-teen (15–17) | 85.3 | 85.9 | 88.0 | 91.7 | 88.6 | 0.7 | 2.4 | 4.2 | *7.5 | <0.001 |
| Late-teen (18–19) | 73.7 | 72.3 | 71.6 | 72.4 | 72.2 | −1.9 | −1.0 | 1.1 | −1.8 | 0.679 |
| **Age at first sex (year)** | | | | | | | | | | |
| <15 | 81.7 | 78.4 | 85.6 | 82.7 | 82.5 | −4.0 | 9.2 | −3.4 | *1.2 | <0.001 |
| 15-19 | 66.8 | 64.2 | 67.1 | 64.6 | 65.4 | −3.9 | 4.5 | −3.7 | −3.3 | 0.007 |
| ≥20 | 12.2 | 11.9 | 8.7 | 7.2 | 9.3 | −2.5 | −26.9 | −17.2 | −41.0 | <0.001 |
| **Contraceptive use** | | | | | | | | | | |
| Never | 67.6 | 60.7 | 63.2 | 61.1 | 62.0 | −10.2 | 4.1 | −3.3 | −9.6 | <0.001 |
| Ever | 53.1 | 41.8 | 42.1 | 45.7 | 44.0 | −21.3 | 0.7 | 8.6 | −13.9 | 0.536 |
| **Media Exposure** | | | | | | | | | | |
| None | 70.3 | 64.5 | 67.2 | 71.4 | 68.5 | −8.3 | 4.2 | 6.3 | *1.6 | <0.001 |
| Exposed | 60.1 | 51.7 | 54.2 | 49.8 | 52.4 | −14.0 | 4.8 | −8.1 | −17.1 | <0.001 |
| **Place of Residence** | | | | | | | | | | |
| Urban | 52.8 | 41.2 | 44.0 | 42.2 | 43.1 | −22.0 | 6.8 | −4.1 | −20.1 | <0.001 |
| Rural | 67.3 | 61.1 | 66.0 | 66.7 | 64.9 | −9.2 | 8.0 | 1.1 | −0.9 | 0.001 |

*(Continued)*

**Table 2.** (Continued)

| Background Characteristics | Prevalence | | | | | Percent changes | | | | |
|---|---|---|---|---|---|---|---|---|---|---|
| | 2003 | 2008 | 2013 | 2018 | Overall | 2003-2008 | 2008-2013 | 2013-2018 | 2003-2018 | p-value |
| **Region** | | | | | | | | | | |
| North Central | 58.7 | 57.7 | 54.6 | 55.6 | 56.0 | −1.7 | −5.4 | 1.8 | −5.3 | 0.019 |
| North East | 77.7 | 72.4 | 69.0 | 72.4 | 71.7 | −6.8 | −4.7 | 4.9 | −6.8 | 0.515 |
| North West | 75.2 | 72.2 | 75.6 | 78.7 | 75.8 | −4.0 | 4.7 | 4.1 | *4.7 | <0.001 |
| South East | 36.1 | 32.8 | 35.5 | 30.8 | 33.1 | −9.1 | 8.2 | −13.2 | −14.7 | <0.001 |
| South South | 56.1 | 44.9 | 43.8 | 39.3 | 43.7 | −20.0 | −2.4 | −10.3 | −29.9 | <0.001 |
| South West | 39.9 | 36.8 | 37.9 | 32.6 | 35.8 | −7.8 | 3.0 | −14.0 | −18.3 | <0.001 |
| **Community SES** | | | | | | | | | | |
| Least disadvantage | 33.5 | 34.2 | 36.5 | 34.8 | 35.0 | 2.1 | 6.7 | −4.7 | 3.9 | 0.074 |
| Low disadvantage | 52.9 | 48.6 | 45.0 | 62.1 | 52.3 | −8.1 | −7.4 | 38.0 | *17.4 | <0.001 |
| Middle disadvantage | 64.9 | 58.7 | 55.4 | 70.9 | 61.7 | −9.6 | −5.6 | 28.0 | *9.2 | <0.001 |
| High disadvantage | 65.7 | 63.5 | 68.2 | 68.6 | 66.7 | −3.3 | 7.4 | 0.6 | *4.4 | <0.001 |
| Most disadvantage | 62.5 | 66.7 | 73.3 | 75.5 | 71.0 | 6.7 | 9.9 | 3.0 | *20.8 | <0.001 |
| ***Total*** | *62.4* | *54.1* | *57.0* | *55.7* | *56.1* | *−13.3* | *5.4* | *−2.3* | *−10.7* | *<0.001* |

* – significant upward trend at p<0.05; SES – socio-economic status

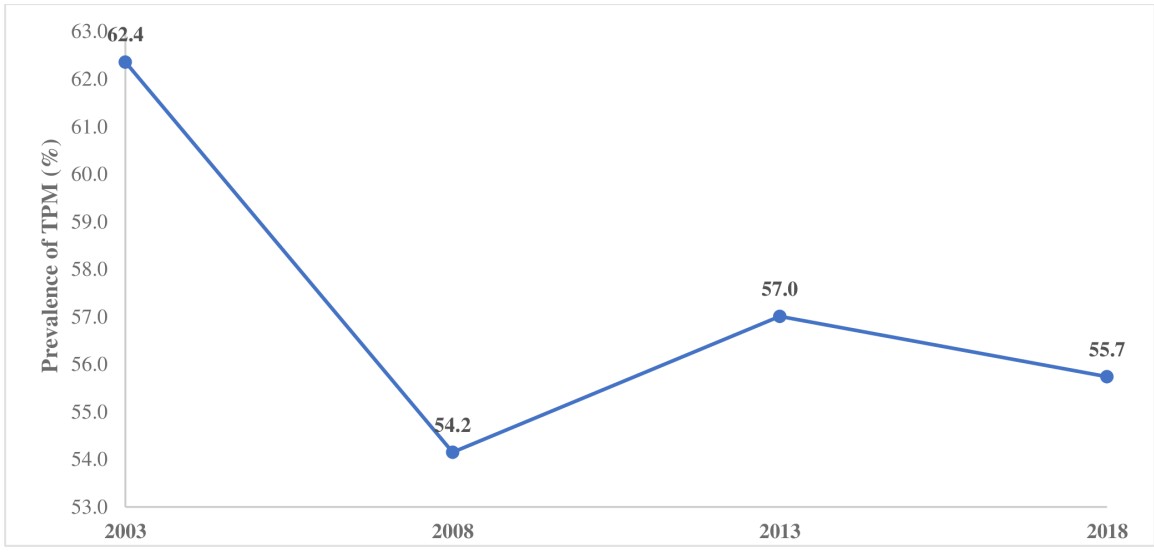

**Fig 1. Percentage distribution of TPM by survey year.**

of the girl child. Such information is necessary as one of the yardsticks for the attainment of SDG 3, particularly related to reducing maternal mortality and preventable newborn deaths in Nigeria and worldwide. Therefore, the study aimed to identify the levels, trends, and drivers of changes in TPM in Nigeria in the last two decades.

In this study, about three-fifths of women had experienced TPM in Nigeria for nearly two decades. The trend changes in having experienced TPM reduced in nearly all the last four consecutive waves of NDHS studied, except for an increase of five percent between 2008 and 2013. Overall, the prevalence of having experienced TPM showed a long-term decreasing

**Table 3. Decomposition of change in odds of teenage pregnancy and motherhood in Nigeria, 2003–2018.**

| Background | Difference due to characteristics (E) | | | Difference due to coefficients (C) | | |
|---|---|---|---|---|---|---|
| Characteristics | Coefficient (s.e) | p-value | % | Coefficient (s.e) | p-value | % |
| **Age (year)** | | | | | | |
| 20-24 | 0.0040 (0.0013) | 0.002 | 6.0 | 0.0112 (0.0060) | 0.063 | 16.9 |
| 25-34 | −0.0002 (0.0002) | 0.204 | −0.4 | 0.0074 (0.0110) | 0.500 | 11.2 |
| 35-49 | Ref | | | | | |
| **Education** | | | | | | |
| No education | 0.0016 (0.0028) | 0.578 | 2.4 | 0.0004 (0.0134) | 0.975 | 0.6 |
| Primary | 0.0058 (0.0018) | 0.001 | 8.7 | 0.0053 (0.0051) | 0.302 | 8.0 |
| Post-primary | Ref | | | | | |
| **Employment** | | | | | | |
| Not working | −0.0034 (0.0015) | 0.021 | −5.1 | −0.0120 (0.0075) | 0.107 | −18.1 |
| Currently working | Ref | | | | | |
| **Wealth Index** | | | | | | |
| Poor | Ref | | | | | |
| Average | −0.0001 (0.0001) | 0.418 | −0.1 | 0.0056 (0.0065) | 0.387 | 8.5 |
| Rich | 0.0006 (0.0007) | 0.367 | 0.9 | 0.0111 (0.0142) | 0.433 | 16.8 |
| **Ethnicity** | | | | | | |
| Hausa/Fulani | −0.0005 (0.0010) | 0.605 | −0.7 | −0.0047 (0.0120) | 0.695 | −7.1 |
| Igbo | 0.0033 (0.0016) | 0.038 | 4.9 | −0.0115 (0.0095) | 0.226 | −17.3 |
| Yoruba | 0.0018 (0.0018) | 0.336 | 2.7 | −0.0090 (0.0078) | 0.247 | −13.6 |
| Others | Ref | | | | | |
| **Religion** | | | | | | |
| Christian | 0.0005 (0.0014) | 0.717 | 0.8 | 0.0022 (0.0404) | 0.956 | 3.4 |
| Muslim | −0.0003 (0.0006) | 0.609 | −0.5 | −0.0170 (0.0462) | 0.713 | −25.7 |
| Others | Ref | | | | | |
| **Timing of marriage (year)** | | | | | | |
| Unmarried at 20 | Ref | | | | | |
| Early-teen (13–14) | 0.0998 (0.0096) | 0.000 | 150.6 | 0.0295 (0.0149) | 0.048 | 44.5 |
| Mid-teen (15–17) | −0.0044 (0.0003) | 0.000 | −6.6 | −0.0129 (0.0090) | 0.154 | −19.4 |
| Late-teen (18–19) | −0.0062 (0.0005) | 0.000 | −9.4 | 0.0044 (0.0040) | 0.275 | 6.7 |
| **Age at first sex (year)** | | | | | | |
| <15 | 0.0146 (0.0071) | 0.041 | 22.0 | −0.0668 (0.0270) | 0.013 | −100.8 |
| 15-19 | −0.0176 (0.0035) | 0.000 | −26.5 | −0.1052 (0.0438) | 0.016 | −158.8 |
| ≥20 | Ref | | | | | |
| **Contraceptive use** | | | | | | |
| Never | Ref | | | | | |
| Ever | 0.0008 (0.0004) | 0.026 | 1.3 | −0.0074 (0.0091) | 0.419 | −11.1 |
| **Media Exposure** | | | | | | |
| None | Ref | | | | | |
| Exposed | 0.0013 (0.0014) | 0.333 | 2.0 | 0.0154 (0.0214) | 0.473 | 23.2 |
| **Place of Residence** | | | | | | |
| Urban | Ref | | | | | |
| Rural | −0.0027 (0.0023) | 0.225 | −4.1 | −0.0097 (0.0130) | 0.459 | −14.6 |
| **Region** | | | | | | |
| North Central | Ref | | | | | |
| North East | 0.0032 (0.0014) | 0.028 | 4.8 | 0.0134 (0.0080) | 0.091 | 20.3 |

*(Continued)*

Table 3. (Continued)

| Background | Difference due to characteristics (E) | | | Difference due to coefficients (C) | | |
|---|---|---|---|---|---|---|
| Characteristics | Coefficient (s.e) | p-value | % | Coefficient (s.e) | p-value | % |
| North West | 0.0001 (0.0004) | 0.728 | 0.2 | −0.0014 (0.0148) | 0.927 | −2.1 |
| South East | −0.0022 (0.0022) | 0.314 | −3.3 | 0.0062 (0.0088) | 0.479 | 9.4 |
| South South | 0.0060 (0.0016) | 0.000 | 9.1 | 0.0128 (0.0061) | 0.035 | 19.3 |
| South West | −0.0051 (0.0027) | 0.061 | −7.7 | 0.0151 (0.0108) | 0.162 | 22.8 |
| Community SES | | | | | | |
| Least disadvantage | Ref | | | | | |
| Low disadvantage | 0.0054 (0.0034) | 0.109 | 8.1 | −0.0205 (0.0137) | 0.135 | −31.0 |
| Middle disadvantage | −0.0013 (0.0049) | 0.789 | −2.0 | −0.0042 (0.0071) | 0.557 | −6.3 |
| High disadvantage | −0.0116 (0.0068) | 0.088 | −17.5 | −0.0107 (0.0083) | 0.193 | −16.2 |
| Most disadvantage | −0.0093 (0.0066) | 0.155 | −14.1 | −0.0093 (0.0074) | 0.205 | −14.1 |
| Constant | | | | 0.1448 (0.1299) | 0.265 | 218.5 |
| Total disparity | 0.0837 (0.0093) | 0.000 | 126.3 | −0.0175 (0.0109) | 0.109 | −26.3 |

Ref – reference category; SES – socio-economic status; s.e - standard error

secular trend, reducing significantly by 10.7% over the last four survey waves. These findings corroborate the outcomes of previous studies in Nigeria and elsewhere in sub-Saharan Africa, where a reduction in teenage pregnancy and adolescent childbearing has been reported [6,35]. However, contrary to our findings, countries like Tanzania, Malawi, and Rwanda have experienced an upsurge in teenage pregnancy [5,36]. The difference could be attributed to the disparity across contextual factors, such as exposure to media, socioeconomic factors, and family dynamics, influencing TPM between Nigeria and these countries [37].

Despite the overall reduction, a significant increasing trend in TPM was observed over the studied period among specific groups of women. These included women who were: unexposed to media, unemployed, Hausa/Fulanis, lacking formal education, married in early- or mid-teens, sexually initiated before age 15, or never used contraceptives. The trend also persisted among women from poor households, the North-West region of Nigeria, or communities with either low, middle, high or most disadvantaged socio-economic status. This finding emphasises the multifaceted and multidimensional factors responsible for TPM. It rarely occurs as the result of a singular factor [19]. To delineate the contribution of each characteristic in terms of compositional or behavioural effects, decomposition analysis is most appropriate for detailing this multidimensionality [33,34].

The observed reduced change in TPM prevalence was largely due to compositional factors, though with nearly 26% nonsignificant increased contribution due to the behavioural effect. Among the compositional factors, a significant contribution to the change in TPM among the studied women was due to age, educational level, employment status, age at sexual initiation, timing of marriage, contraceptive use, ethnicity, and region. Similar studies previously conducted in Africa have documented these important compositional factors [5,17,35]. However, in a similar study conducted in Rwanda, Uwizeye et al reported a rapid increase in teenage pregnancy in the last two decades [36]. This is despite the Rwandan government's interventions through improved sex education and defilement laws to punish men who impregnate underage girls [38]. This finding suggests that government interventions alone are inadequate to reduce the burden of TPM, emphasizing the critical role of contextual factors in influencing its occurrence [19,23,39].

In this study, the reduced composition of respondents aged <25 years had a significant impact on reducing the trend of TPM. It also appears the shift in age was towards a positive change in behaviour of younger respondents, which exerted a decreasing trend on the prevalence of TPM, although marginally, not statistically significant. Saliently, a general shift and

decrease in the composition of teenagers marrying before attaining age 15 years explained the highest percent reduction in TPM between 2003 and 2018. Ironically, there was an increase in the composition of women married in mid- and late-teens, which was associated with an increasing trend in TPM.

Generally, child marriage is positively associated with early sexual intercourse, and early sexual initiation is one of the predisposing factors for early pregnancy and motherhood [5]. The compositional effect of age at sexual initiation was positive for those who had sexual initiation before age 15 years, and negative for those aged 15–19 years. Overall, the negative endowment effect of the increase in the share of teens aged 15–19 years offset the positive endowment effect of those aged <15 years. This increasing trend in TPM is further compounded by the change in the behaviour of teenagers, which significantly induced an increasing trend in TPM. Notably, shifts in behaviour were more important than changes in the structure in explaining this increase. If this is left unaddressed, it could increase to chains of inter-generational poverty and low educational attainment [22]. There is a need to ensure that the girl-child is reoriented and equipped with a timely and qualitative education, including accessible, affordable, and socially acceptable contraceptives.

In a UNICEF report, one in four adolescent girls aged 15–19 years who want to avoid pregnancy are currently not using a modern method [40]. Various reasons may account for the poor use of contraceptives among teenagers. Some of these are lack of information and knowledge about contraceptives and their uses, and inability to procure contraceptives either because of lack of access or money to obtain them. In addition, sexual and reproductive health services where such information is given to this age group are not readily available in most places. And where sexual and reproductive health services exist, culturally entrenched myths and misconceptions about contraceptive use are likely to prevent the uptake of the services, especially for teens [2]. A report says meeting the unmet need for contraception of girls aged 15–19 years would reduce unintended pregnancies by six million annually, averting 2.1 million unplanned births, 3.2 million abortions, and 5600 maternal deaths [40].

Not surprisingly, in both high- and low-income countries, including Nigeria, early sexual initiation is a significant factor in the occurrence of teenage pregnancy [41]. Early sexual debut was reported as a risk factor for TPM among adolescents in Johannesburg and Baltimore's urban disadvantaged settings across five cities [42]. Factors that predispose the girl child to early sexual initiation may include unmet needs for contraception, poverty, peer pressure, and early marriage [16,42,43]. The outcome of this may include unplanned pregnancy, sexually transmitted infections, and poor reproductive health outcomes, among others.

Reduction in the composition of women with primary higher education, compared with women with no formal education, contributed 9% to the reduction in TPM over the study period. This suggests the protective effects of education on reducing TPM prevalence between 2003 and 2018. A qualitative study by Esan et al on the causes, enablers and solutions to teenage pregnancy in a southwestern state in Nigeria documented lack of education as an enabler of teenage pregnancy [39]. Other studies have also found higher rates of teenage pregnancy among those with a low level of education [2,41]. Studies from South Africa, Ghana, and other regions of the world also reported a high occurrence of TPM among teenagers with no education [44–46]. Morh and Sharma, in a systematic review on teenage pregnancy in low-income countries, reported a significant association between low educational attainment and the occurrence of teenage pregnancy [47].

Education plays a vital role in delaying age at marriage, empowering the girl child with career aspirations [48], and knowledge about health and well-being, particularly sexual and reproductive health, sex education, and contraceptive use. It appears the shift in contraception was towards an increase in the contribution of respondents who have ever used contraceptives, which exerted a decreasing trend on the prevalence of TPM. Despite high awareness of contraceptives among the respondents, their uptake was low. Similar studies reported poor contraceptive use and high TPM among teenage girls in Nigeria, Africa, and other countries [22,43,49]. This contrasts with a study by Habito et al among teenagers in the Philippines, which reported that ever-used contraception was positively associated with teenage pregnancy. This may, however, suggest contraceptive use after a prior pregnancy [50].

Notably, this study demonstrated a shift and decrease in the composition of unemployed women, which was associated with an increasing trend in TPM. A possible explanation could be that employment often provides women with the needed

income and confidence to access and utilise sexual reproductive services. However, teens who lived in poor households and communities with the most disadvantaged socio-economic status had a rise in percent change of TPM. This is an offshoot of poverty and poor education. Studies in other African countries have reported high rates of TPM among teenagers from poor households [19,23]. Poverty makes the girl child engage in financial struggles, and as a way of escape, she engages in transactional sex, possibly with multiple sexual partners. Other studies have documented child marriages among girls in the poorest wealth quintile, basically for financial gain and escape from economic hardship [16,51]. These increase the vulnerability of the girl child to TPM.

In addition, the region of residence relatively contributed to the change in TPM to the extent that the highest percentage change to the reduction in TPM was observed in the South-South compared to other regions, especially the northern regions. Some other studies reported similar findings, with a high occurrence of teen motherhood in the northern part of Nigeria [16,17]. Socio-cultural practices and socio-economic disparities among ethnic groups could have accounted for this variation. Poverty, early marriage, and low education and literacy levels are common in the northern regions [15,52]. The regions embrace the culture of early marriage, which is mostly entrenched among people with low socio-economic status for financial gain.

The decline in the composition of North-East and South-South residents which was associated with reducing TPM may be linked partly to the instability and conflict, such as Boko Haram insurgence (North-East) and militancy (South-South) [53]. Besides the decline in the proportion of South-South residents, behaviour changes among the residents contributed significantly to the decrease in TPM.

## Limitations

The present study, however, acknowledges some limitations. First, the cross-sectional study design limits the potential of making causal inferences. Second, the actual burden of TPM might have been underestimated due to women's self-reported data without verification, which could be a potential avenue of recall bias. Third, the respondents' preference for socially desirable responses might have affected the data. Additionally, the study limited the choices of independent variables. Nonetheless, the study demonstrated some strengths. First, this study highlighted changes in TPM between 2003 and 2018, and decomposed factors contributing to the changes in Nigeria. The study used a robust decomposition method appropriate for handling dichotomous outcomes, which has been adjudged to be an effective tool in detailing multi-dimensionality [31,33,34]. In addition, the study followed the pregnancy history of adults during their teenage years to estimate TPM levels, and the use of large and nationally representative four consecutive waves of dataset allows the study findings to be generalisable across the country.

## Conclusions

This study showed a reduction in TPM in nearly the past two decades. Nonetheless, TPM was most prevalent among women who lived in the low- to most-disadvantaged socio-economic community, followed by those who married before attaining age 18 years, resided in poor households or North-West, and were uneducated, unemployed, or Hausas/Fulanis. The identified important contributors to an increasing trend in TPM include unemployment and early initiation of sexual intercourse. Therefore, there is an urgent need to strengthen sexual and reproductive health education among this population. In addition, concerted intervention from individuals, families, communities, governments, and stakeholders is required to synergise efforts, including actionable policy dedicated primarily towards alleviating TPM in Nigeria.

## Acknowledgments

The authors appreciate The DHS program, ICF International, USA for providing us with access to use the data.

## Author contributions

**Conceptualization:** Mobolaji M. Salawu, Rotimi Felix Afolabi.

**Data curation:** Rotimi Felix Afolabi.

**Formal analysis:** Rotimi Felix Afolabi.

**Investigation:** Rotimi Felix Afolabi.

**Methodology:** Rotimi Felix Afolabi.

**Software:** Rotimi Felix Afolabi.

**Supervision:** Mobolaji M. Salawu, Ayo Stephen Adebowale.

**Validation:** Mobolaji M. Salawu, Rotimi Felix Afolabi, Ayo Stephen Adebowale.

**Visualization:** Rotimi Felix Afolabi, Ayo Stephen Adebowale.

**Writing – original draft:** Mobolaji M. Salawu, Rotimi Felix Afolabi.

**Writing – review & editing:** Mobolaji M. Salawu, Rotimi Felix Afolabi, Ayo Stephen Adebowale, Martin Enock Palamuleni.

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
