## [Decision Letter · Decision Letter 0]

24 Mar 2025

Dear Dr. Afolabi,

Thank you for submitting your manuscript to PLOS ONE. After careful consideration, we feel that it has merit but does not fully meet PLOS ONE’s publication criteria as it currently stands. Therefore, we invite you to submit a revised version of the manuscript that addresses the points raised during the review process.

**ACADEMIC EDITOR:**

The manuscript provides a well-structured analysis of trends and factors associated with teenage pregnancy and motherhood (TPM) in Nigeria over 15 years. In addition to expert reveweers' feedback, below are some comments for the authors to consider:

Ensure consistency in terminologyEnsure sentences make complete sense. For example, ‘This has contributed …’ (line 66). What does ‘this’ refer to? Please, extend this correction to all relevant sections of the manuscript.‘Research has revealed that achieve a reduction in the global population growth rate …’ (lines 90 – 91), is not clear.Some sentences are lengthy and could be restructured for clarity.Avoid the use of emotive words e.g. ‘serious’Where appropriate, use ‘childbirth’ instead of ‘delivery’How did previous studies underestimate TPM in Nigeria? It would be helpful to briefly highlight why previous cross-sectional studies may have underestimated TPM burden in Nigeria.Line 101:  how will the study contribute to attaining SDG 3?A brief explanation of why the decomposition method is particularly suited for TPM analysis would strengthen the rationale, especially, in comparison to previous studies.The manuscript highlights important intervention points, but it would benefit from more specific policy recommendations.

We look forward to receiving your revised manuscript.

Kind regards,

Emmanuel O Adewuyi, BPharm, MPH, PhD

Academic Editor

PLOS ONE

Reviewers' comments:

Reviewer's Responses to Questions

**Comments to the Author**

1. Is the manuscript technically sound, and do the data support the conclusions?

Reviewer #1: Yes

Reviewer #2: Yes

Reviewer #3: Yes

Reviewer #4: Yes

2. Has the statistical analysis been performed appropriately and rigorously?

Reviewer #1: Yes

Reviewer #2: Yes

Reviewer #3: Yes

Reviewer #4: Yes

3. Have the authors made all data underlying the findings in their manuscript fully available?

Reviewer #1: Yes

Reviewer #2: Yes

Reviewer #3: Yes

Reviewer #4: Yes

4. Is the manuscript presented in an intelligible fashion and written in standard English?

Reviewer #1: Yes

Reviewer #2: Yes

Reviewer #3: No

Reviewer #4: Yes

Reviewer #1: The paper provides the valuable insights into the dynamics of teenage pregnancy in Nigeria and also methodologically sound. in order to improve the manuscript further, the authors should address the following points

1. Teenage pregnancy and motherhood(TPM) should be operationally defined

2.Authors should acknowledge the limitations related to self-reported data and cross sectional nature of the study design

Reviewer #2: Add inclusion and exclusion criteria details, why age range is till 49, what about abouve that age.

Also, authours can add acxess to health case dacility or primary health education, role of family planning services in that area, it should be discussed or assesed through data. Highly appreciated the aim of the study to assess the trends and was also conpare to some extend but what is the outcome, what is next, what is the importance of this trend data, will it be compare with governemnt interventions ?, authors can further add the cimparision disscusion between two stat3s where one has highest numbers of TPM and other has reducing trends in terms of elaborating what majors that other srate has raken to reduce the numbers.

Elaborate the aim in the abstract so that readers can get the main important goal of this study.

Reviewer #3: The paper is overall ineresting, however some comments are necessary:

- In the Abstract I would rather speak of “the limited number of studies”

- lines 105-115 all goes to introduction, two lines maximum can be kept to provide the specific context of the study maybe also changing the paragraph title to include the word "context" although it is not mandatory

- lines 120-1212 eliminate this unnecessary phrase

- In general, the English form needs revising, eliminate spelling mistakes like 20013, 247 and other lines, please choose better vocabulary for scientific purposes.

- Please clarify in the methods why the specific focus on women reporting and reported at least one childbirth, pregnancy termination, or 32 stillbirth before attaining age 20 and not just considering all women experiencing pregnancy in that age class, considering you cite in line 143 “irrespective of whether the pregnancy resulted in childbirth or was voluntarily or involuntarily terminated”. This is a major issue

- State clearly in the methods the age class as there seems to be inconsistency between abstract and manuscript methods. Furthermore, detail on age class needs to go the results section

- You mention “The detail of the ethical approval was reported by Measure DHS”. Please state exactly where it was reported and a reference to find this approval

- The Discussion section is too long, please sum up and leave out inessential concepts

- With regards to lines 315-320, a major role is also played by public health policies and programs in place. Please expand on different public health approaches of these countries so as to add complexity to the understanding of observed differences among these countries

Reviewer #4: The study offers valuable insights into the trends and determinants of teenage pregnancy and motherhood in Nigeria. There are some comments to improve your manuscript:

1. What is the meaning of "media exposure"?

2. How was household wealth status measure?

3. The lines 173-176 are not clear. The method used to define community SES needs more detailed explanation.

4. How was the test for asterisk number in table 2 conducted? Please mention the name of the test and explain the difference between the tests with the reported p-values on the last column of Table 2.

5. Some percentage in lines 290-302 do not exactly match the percentage in Table 3. This may be due to rounding. However, it is suggested that the percentage should be equal to those in Table 3. For example, report 134.7% instead of 134.7% or 5.6 % instead of 5.7%.

6. In Table 3, please report the standard error of the coefficient or the 95% confidence intervals.

**Do you want your identity to be public for this peer review?** For information about this choice, including consent withdrawal, please see our Privacy Policy

Reviewer #1: No

Reviewer #2: **Yes: ** Zaibunissa

Reviewer #3: No

Reviewer #4: **Yes: ** Farzane Ahmadi

---

## [Author Response · Author response to Decision Letter 1]

23 Apr 2025

We appreciate the efforts of all the reviewers for the thorough review of our manuscript. The point-by-point responses to all comments have been provided in the attached Point-by-point responses to the reviewers' comments file

---

## [Decision Letter · Decision Letter 1]

19 May 2025

Trend and decomposition analysis of factors influencing teenage pregnancy and motherhood in Nigeria, 2003 – 2018.

PONE-D-24-38929R1

Dear Dr. Afolabi,

We’re pleased to inform you that your manuscript has been judged scientifically suitable for publication and will be formally accepted for publication once it meets all outstanding technical requirements.

Kind regards,

Emmanuel O Adewuyi, BPharm, MPH, PhD

Academic Editor

PLOS ONE

Additional Editor Comments (optional):

Reviewers' comments:

Reviewer's Responses to Questions

**Comments to the Author**

Reviewer #1: All comments have been addressed

Reviewer #4: All comments have been addressed

2. Is the manuscript technically sound, and do the data support the conclusions?

Reviewer #1: Yes

Reviewer #4: Yes

3. Has the statistical analysis been performed appropriately and rigorously?

Reviewer #1: Yes

Reviewer #4: Yes

4. Have the authors made all data underlying the findings in their manuscript fully available?

Reviewer #1: Yes

Reviewer #4: Yes

5. Is the manuscript presented in an intelligible fashion and written in standard English?

Reviewer #1: Yes

Reviewer #4: Yes

Reviewer #1: (No Response)

Reviewer #4: (No Response)

**Do you want your identity to be public for this peer review?** For information about this choice, including consent withdrawal, please see our Privacy Policy

Reviewer #1: **Yes: ** Zeleke Dutamo Agde

Reviewer #4: **Yes: ** Farzane Ahmadi

---

## [Editor Report · Acceptance letter]

PONE-D-24-38929R1

PLOS ONE

Dear Dr. Afolabi,

I'm pleased to inform you that your manuscript has been deemed suitable for publication in PLOS ONE. Congratulations! Your manuscript is now being handed over to our production team.

Kind regards,

on behalf of

Dr. Emmanuel O Adewuyi

Academic Editor

PLOS ONE